

# Risk factors for high CAD-RADS scoring in CAD patients revealed by machine learning methods: a retrospective study

Yueli Dai[1,*], Chenyu Ouyang[2,*], Guanghua Luo[2], Yi Cao[1], Jianchun Peng[1], Anbo Gao[3,4,5] and Hong Zhou[2]

[1] Department of Radiology, The Second Affiliated Hospital, Hengyang Medical School, University of South China, Hengyang, Hunan, China

[2] Department of Radiology, The First Affiliated Hospital, Hengyang Medical School, University of South China, Hengyang, Hunan, China

[3] Clinical Research Institute, The Second Affiliated Hospital, Hengyang Medical School, University of South China, Hengyang, Hunan, China

[4] Department of Cardiovascular Medicine, The Second Affiliated Hospital, Hengyang Medical School, University of South China, Hengyang, Hunan, China

[5] Key Laboratory of Heart Failure Prevention & Treatment of Hengyang, Clinical Medicine Research Center of Arteriosclerotic Disease of Hunan Province, Hengyang, Hunan, China

[*] These authors contributed equally to this work.

Corresponding authors
Anbo Gao, 513904562@qq.com
Hong Zhou, zhouhong@msn.cn

## ABSTRACT

**Objective**. This study aimed to investigate a variety of machine learning (ML) methods to predict the association between cardiovascular risk factors and coronary artery disease-reporting and data system (CAD-RADS) scores.

**Methods**. This is a retrospective cohort study. Demographical, cardiovascular risk factors and coronary CT angiography (CCTA) characteristics of the patients were obtained. Coronary artery disease (CAD) was evaluated using CAD-RADS score. The stenosis severity component of the CAD-RADS was stratified into two groups: CAD-RADS score 0-2 group and CAD-RADS score 3–5 group. CAD-RADS scores were predicted with random forest (RF), k-nearest neighbors (KNN), support vector machines (SVM), neural network (NN), decision tree classification (DTC) and linear discriminant analysis (LDA). Prediction sensitivity, specificity, accuracy and area under the curve (AUC) were calculated. Feature importance analysis was utilized to find the most important predictors.

**Results**. A total of 442 CAD patients with CCTA examinations were included in this study. 234 (52.9%) subjects were CAD-RADS score 0–2 group and 208 (47.1%) were CAD-RADS score 3–5 group. CAD-RADS score 3-5 group had a high prevalence of hypertension (66.8%), hyperlipidemia (50%) and diabetes mellitus (DM) (35.1%). Age, systolic blood pressure (SBP), mean arterial pressure, pulse pressure, pulse pressure index, plasma fibrinogen, uric acid and blood urea nitrogen were significantly higher ($p < 0.001$), and high-density lipoprotein (HDL-C) lower ($p < 0.001$) in CAD-RADS score 3–5 group compared to the CAD-RADS score 0–2 group. Nineteen features were chosen to train the models. RF (AUC = 0.832) and LDA (AUC = 0.81) outperformed SVM (AUC = 0.772), NN (AUC = 0.773), DTC (AUC = 0.682), KNN (AUC = 0.707). Feature importance analysis indicated that plasma fibrinogen, age and DM contributed most to CAD-RADS scores.

**Conclusion**. ML algorithms are capable of predicting the correlation between cardiovascular risk factors and CAD-RADS scores with high accuracy.

# INTRODUCTION

CAD is the leading cause of morbidity and mortality worldwide (*Akella & Akella, 2021*; *Popa et al., 2020*; *Saharan et al., 2021*) and is one of the major contributors to healthcare costs in China. The pathogenesis of CAD is complex and is affected by a variety of risk factors, with atherosclerosis being the most common underlying cause of cardiovascular diseases (*Popa et al., 2020*). Multiple conventional risk factors augment the atherosclerotic process, including age,sex,smoking, hypertension, hyperlipidemia, DM, hyperuricemia, coagulation abnormalities, obesity, insulin resistance, C-reactive protein levels, plasma fibrinogen, and others (*Giacco & Brownlee, 2010*; *Kazemian et al., 2020*; *Shariatnia et al., 2022*; *Song et al., 2015*; *Tsai, Chiang & Huang, 2020*; *Velusamy & Ramasamy, 2021*; *Williams et al., 2018*; *Yang et al., 2018*). It is indispensable to comprehend and properly calculate the etiological contribution of these risk factors to devise and improve preventive tactics for CAD. In 2016, the Society of Cardiovascular Computed Tomography (SCCT), the American College of Radiology (ACR), and the North American Society for Cardiovascular Imaging (NASCI) published the CAD-RADS, which is a new standardized method to assess CAD using CCTA (*Rubinshtein & Hamdan, 2020*). Although there have been numerous studies on CAD risk prediction, studies involving the application of CAD-RADS on traditional risk factors on the Chinese population evaluated by CCTA remain understudied, and the impact of CAD-RADS management and outcome is still unknown (*Foldyna et al., 2018*), while risk assessment is crucial for the reduction of the worldwide burden of CAD.

Machine learning (ML) have been developed to predict outcomes in cardiovascular disease and have the potential to provide useful insights for cardiovascular medicine systems (*Khalaji et al., 2022*; *Li et al., 2022*). ML accommodates most artificial intelligence (AI) technologies in the medical research setting and includes various algorithms for prediction and classification tasks that perform well on complex big data (*Kagiyama et al., 2019*). These algorithms have emerged as valuable tools for predicting patient outcomes based on pertinent feature characteristics variables and have already been applied to identify unknown CAD risk factors, automate imaging interpretation, and enhance clinical decision-making, thus facilitating precision medicine (*Huang et al., 2022*; *Panteris et al., 2022*; *Saravi et al., 2022*). Some of the most widely used mathematical methods for predictions are discriminant analysis, logistic regression, neural networks, and classification and regression trees (*Shariatnia et al., 2022*). The strongest predictors can be selected to train the system to predict outcomes using supervised learning (*Khalaji et al., 2022*). Although ML had been applied in literature to predict CAD-RADS scores, few studies had evaluated commonly used clinical risk factors in predicting these scores

(*Muscogiuri et al., 2020*). This study hypothesizes that ML algorithms have the potential to accurately predict CAD-RADS scores based on the most significant cardiovascular risk factors.

## MATERIALS & METHODS

### Study Design and data collection

Participants were selected from patients who visited the cardiology department of the First or the Second Affiliated Hospitals of University of South China between October 2017 and December 2022. Demographic and clinical data were collected retrospectively, and information on clinical risk factors was obtained, the time interval between the collection of clinical risk factors and CCTA data was two weeks. Out of the 579 participants who underwent a CCTA scan, 442 subjects were included in the study after excluding those with missing or unsatisfactory CCTA data for analysis ($n = 45$), incomplete basic clinical information ($n = 38$), and a history of bypass surgery or percutaneous coronary intervention (PCI) ($n = 54$) (Fig. 1). The study was approved by the human ethics review board of University of South China (2022020587), and all patients provided written informed consent. Inclusion criteria for the study were as follows: all patients had a free heart rate and cardiac rhythm variation of ≤5 beats/min and no obvious contraindications. Exclusion criteria were a history of valvular heart disease, bypass surgery or PCI, severe arrhythmia, and failure to cooperate during inspection. All coronary segments with a diameter greater than 1.5 mm were evaluated according to the Expert Consensus Document (*Cury et al., 2016*).

### Cardiovascular risk assessment

Demographic variables and traditional CAD risk factors included age, gender, SBP, diastolic blood pressure, mean arterial pressure, pulse pressure, pulse pressure index, hypertension, DM, smoking status, hyperlipidemia, total cholesterol, triglycerides, HDL-C, low-density lipoprotein cholesterol (LDL-C), uric acid, plasma fibrinogen, blood creatinine, and blood urea nitrogen. Patients who were smokers at the time of analysis were classified as current smokers. Hypertension is defined as SBP values ≥140 mmHg and/or diastolic blood pressure values ≥90 mmHg or use of antihypertensive medication (*Williams et al., 2018*). DM was defined as fasting serum glucose ≥126 mg/dL (7.0 mmol/L),or 2-hour values in the oral glucose tolerance test ≥200 mg/dL (11.1 mmol/L),or hemoglobin A1c level ≥6.5%. Hyperlipidemia was defined as fasting serum total cholesterol level ≥2.3 mmol/L (220 mg/dL) and/or fasting serum triglyceride level ≥150 mg/dL and/or the use of antihyperlipidemic agents. High LDL-C is defined as LDL-C ≥2.6 mmol/L (100 mg/dL) for the first time, while low HDL-C was HDL-C<1.0 mmol/L (40 mg/dL) (*Zhou et al., 2022*).

### CCTA scan protocol

All CCTA scans were performed using two 256-slice multidetector CT scanners (Brilliance iCT 256 from Philips and SOMATOM Definition Flash CT from Siemens). The scanning parameters were as follows: tube voltage of 120 kV, 800 mAs, slices/collimation of 128/0.625 mm, gantry rotation time of 330 ms, pitch of 0.2, effective slice thickness of 0.9 mm, and

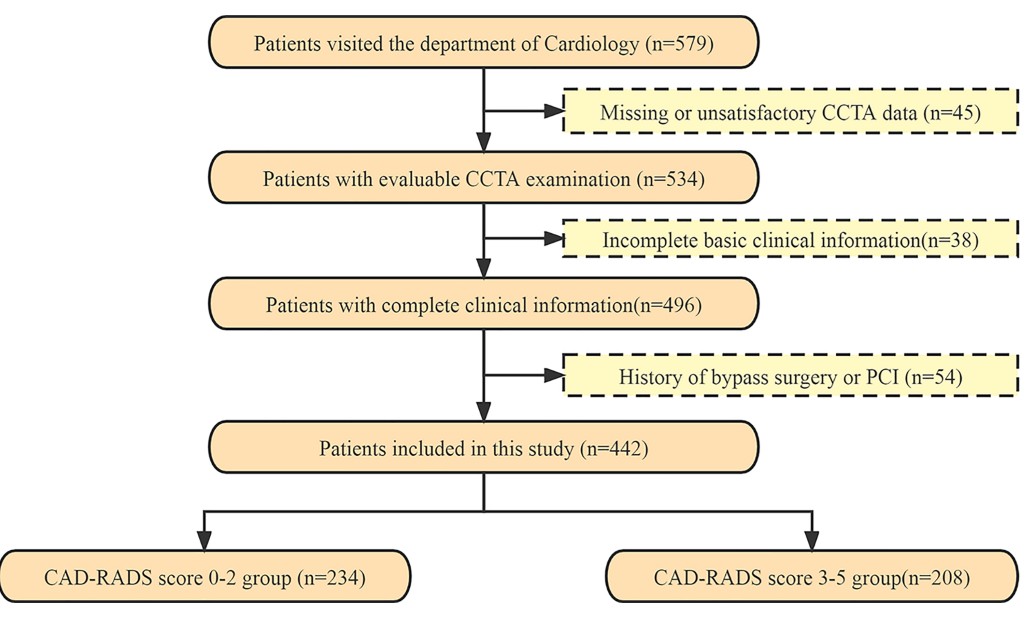

**Figure 1** **The flowchart of patients recruited in this study.** CCTA, coronary computed tomography angiography; PCI, percutaneous coronary intervention; CAD-RADS, Coronary Artery Disease-Reporting and Data System.

reconstruction increment of 0.45 mm. Patients with a heart rate > 80 beats/min were given oral beta-blockers 1 h prior to the examination. All patients received 0.5 mg of sublingual nitroglycerin for coronary vasodilatation. A bolus of 1.5 ml/kg of iodinated contrast medium was administered intravenously at a rate of 5 ml/s, followed by 40 ml of saline injected at the same rate. After acquisition, the images were processed using artificial intelligence (AI) software (V2.4.2; Shukun, Beijing, China).

## Rating of CAD-RADS score

(1) Coronary stenosis severity was scored as follows: CAD-RADS 0 indicated no visible stenosis, with a degree of maximal coronary stenosis of 0%; CAD-RADS 1 indicated minimal stenosis of 1–24%; CAD-RADS 2 indicated mild stenosis of 25–49%; CAD-RADS 3 indicated moderate stenosis of 50–69%; CAD-RADS 4 indicated severe stenosis, including two groups: CAD-RADS 4A-70–99%; CAD-RADS 4B-Left main > 50% or 3-vessel obstructive disease; and CAD-RADS 5 indicated 100% total occlusion (*Cury et al., 2016*; *Foldyna et al., 2018*; *Laggoune et al., 2019*; *Popa et al., 2020*).

(2) The stenosis severity component of CAD-RADS was stratified into two groups for uniformity and sample size based on previously published methods: CAD-RADS score 0-2 group and CAD-RADS score 3–5 group (*Laggoune et al., 2019*; *Popa et al., 2020*). The CAD-RADS scores were generated by AI software (V2.4.2, Shukun). Figure 2 showed the different degrees of coronary artery stenosis in CCTA images.

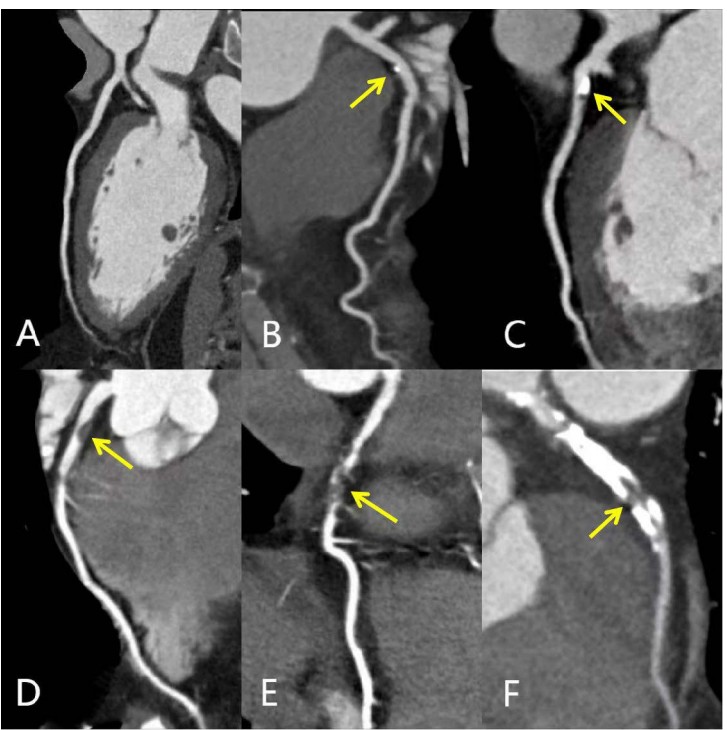

**Figure 2  MPR images of different degrees of coronary artery stenosis.** MPR images showing different degrees of coronary artery stenosis (yellow arrows): (A) Normal LAD without any plaque or stenosis (CAD-RADS 0); (B) minimal calcified plaque in the proximal LAD with minimal luminal narrowing < 25% (CAD-RADS 1); (C) predominantly calcified plaque in the proximal LAD with 25%–49% diameter stenosis (CAD- RADS 2); (D) non-calcified plaque in the proximal LAD with 50%–69% diameter stenosis (CAD- RADS 3); (E) segmental non-calcified plaque in the mid LAD with 70%–99% diameter stenosis (CAD- RADS 4); (F) total occlusion of proximal and mid LAD; calcified and non-calcified mixed plaques support the diagnosis of chronic total occlusion (CAD- RADS 5). CAD-RADS, Coronary Artery Disease-Reporting and Data System; MPR, multiplanar reconstruction; LAD, left anterior descending artery.

## Test/train split and feature selection

The study population was randomly assigned to the training cohort, which comprised 70% of the patients, and the test cohort, which comprised 30% of the sample, in order to validate the predictive models (*Gao et al., 2015*; *Khalaji et al., 2022*). The training dataset was used to train the model, which learned from the data in this dataset. The test dataset was then used to provide an unbiased evaluation of the final model fit to the training dataset (*Akella & Akella, 2021*). Feature selection was performed using a technique known as "information gain attribute ranking" (*Motwani et al., 2017*), the most significant predictors were obtained from the random forest (RF) model prediction in the training data using 10-fold cross-validation (*Khalaji et al., 2022*). The dataset was partitioned into ten distinct subsets, with nine of them designated for training and one for evaluation. This process was repeated ten times using ten different but overlapping sets for training and testing.

## Model development and performance evaluation

To develop predictive models, we used six ML methods: random forest (RF), support vector machine (SVM), neural network (NN), k-nearest neighbor (KNN), decision tree classification (DTC) and linear discriminant analysis (LDA).

All models were implemented using the statistical software package R and JASP and designed using k-fold ($k = 10$) cross-validation. We tuned the parameters for each model using the grid search method to increase the prediction accuracy. The training set was used to learn the ML parameters, while the test set was used for standard evaluation metrics. Each model was trained and tested for CAD-RADS scores.

We evaluated the performance of the ML methods using the following indices: (1) sensitivity, (2) specificity, (3) accuracy of prediction, (4) area under the receiver operating characteristics curve (ROC-AUC), which plots true positive against false positive rate (*Khalaji et al., 2022*; *Liu et al., 2021*; *Shariatnia et al., 2022*). As AUC is a measure of discrimination independent of the threshold, we chose it as the primary index to compare the performance of the models. AUC was interpreted as follows: AUC $\geq 0.9$, outstanding discrimination; $0.8 \leq$ AUC $< 0.9$, excellent discrimination; $0.7 \leq$ AUC $< 0.8$, acceptable/fair discrimination; $0.6 \leq$ AUC $< 0.7$, poor discrimination; AUC $< 0.6$, no discrimination (*Khalaji et al., 2022*).

## Statistical analysis

Statistical analysis was performed using the SSPS software (V25.0; SPSS INc., Chicago, IL, USA). Baseline characteristics are presented as mean $\pm$ standard deviation (SD) or frequencies and percentages. Categorical variables were compared using the chi-square or Fisher's exact tests, while continuous variables were analyzed with independent samples *t*-test. Prior to analysis, we assessed the normality of data distributions and homogeneity of continuous variables.Whenever the distribution of continuous data was not normal, the Mann–Whitney U-test was used for comparison, and results were presented as median (interquartile range, IQR). A *p* value< 0.05 was considered statistically significant. Six models were employed to utilize the statistical software package R and JASP, and their performance was subsequently compared to determine the optimal selection classifier for identifying high risk factors in predicting CAD-RADS scores.

# RESULTS

## Baseline characteristics of the study population

A total of 442 CAD patients were included in this cohort, with 268 (60.6%) males and 174 (39.4%) females. The median age was 63 years, with the lowest and highest ages being 18 and 88 years, respectively. Among the entire cohort, there was a high prevalence of patients with hypertension (51.6%) and hyperlipidemia (42.8%). One hundred and forty-two (32.1%) people had a history of smoking and 91 (20.6%) had DM. All subjects were divided into two groups based on CAD-RADS scores: 234 (52.9%) subjects were CAD-RADS score 0–2 group and 208 (47.1%) were CAD-RADS score 3–5 group. The CAD-RADS score 3–5 group had a higher prevalence of hypertension (66.8%), hyperlipidemia (50%), and DM (35.1%). Age, SBP, mean arterial pressure, pulse pressure, pulse pressure index, plasma fibrinogen,

**Table 1   Univariate analysis for the association between cardiovascular risk factors and CAD classified using CAD- RADS categories.** Data are presented as mean ± S.D, or median (IQR), or number (%).

| | Entire cohort ($n = 442$) | CAD-RADS score 0-2 ($n = 234$) | CAD-RADS score 3-5 ($n = 208$) | statistic | P value |
|---|---|---|---|---|---|
| Age, median, (IQR), y | 63(55–70) | 59(52–66.25) | 67(60–75.75) | 15434.5 | <0.001 |
| Gender (men, %) | 268 (60.6%) | 122(52.1%) | 146(70.2%) | 19.991 | <0.001 |
| Hypertension (%) | 228(51.6%) | 89(38%) | 139(66.8%) | 36.553 | <0.001 |
| Systolic blood pressure, median, (IQR), mmHg | 130(120–150) | 124(114.75–149) | 140(124–154.75) | 17652 | <0.001 |
| Diastolic blood pressure, median, (IQR) mmHg | 80(70–90) | 80(70–88) | 80(70–90) | 21753 | 0.052 |
| Mean arterial pressure, median, (IQR) mmHg | 96.668(86.67–109.5) | 93.33(86.67–106.67) | 100(88.75–113.08) | 19320 | <0.001 |
| Pulse pressure, median, (IQR) mmHg | 50.5(41.75–65) | 50(40–60) | 57.5(49.25–70) | 16698 | <0.001 |
| Pulse pressure index | 0.402 ± 0.076 | 0.388 ± 0.072 | 0.419 ± 0.078 | −4.34 | <0.001 |
| Diabetes mellitus (%) | 91 (20.6%) | 18 (7.7%) | 73 (35.1%) | 152.941 | <0.001 |
| Smoking (%) | 142 (32.1%) | 49 (20.9%) | 93 (44.7%) | 56.48 | <0.001 |
| Hyperlipidemia (%) | 189 (42.8%) | 85 (36.3%) | 104 (50%) | 9.267 | 0.002 |
| Total cholesterol, median, (IQR) mmol/L | 4.56(3.818–5.293) | 4.6(3.898–5.37) | 4.48(3.713–5.208) | 25831 | 0.265 |
| Triglycerides, median, (IQR) mmol/L | 1.425(0.98–2.255) | 1.35(0.938–1.965) | 1.535(1.045–2.445) | 21231 | 0.021 |
| LDL-C (mmol/L) | 2.694 ± 1.013 | 2.668 ± 0.884 | 2.723 ± 1.143 | −0.565 | 0.572 |
| HDL-C (mmol/L) | 1.310 ± 0.372 | 1.366 ± 0.378 | 1.246 ± 0.355 | 3.447 | <0.001 |
| Plasma fibrinogen, median, (IQR) g/L | 2.695(2.14–3.68) | 2.52(2.048–3.19) | 3.35(2.365–4.625) | 14863 | <0.001 |
| Uric acid, median, (IQR), $\mu$mol/L | 321.65(261.75–396.5) | 305.1(247.975–369) | 341(283.25–441) | 18213.5 | <0.001 |
| Serum creatinine, median, (IQR), $\mu$mol/L | 81(66.5–96) | 79.2(64.05–91) | 83.3(68.5–102.925) | 20385.5 | 0.003 |
| Blood urea nitrogen, median, (IQR), mmol/L | 5.3(4.3–6.5) | 5.045(4.09–6.133) | 5.7(4.57–6.975) | 19288.5 | <0.001 |

**Notes.**

SBP, Systolic blood pressure; DBP, Diastolic blood pressure; LDL-C, low-density lipoprotein cholesterol; HDL-C, high-density lipoprotein cholesterol; BUN, Blood urea nitrogen; IQR, interquartile range.

Chisquare tests were performed on gender, hypertension, diabetes, smoking, and hyperlipidemia. Pulse pressure index and HDL-C were tested by Student test. Age, systolic blood pressure, diastolic blood pressure, mean arterial pressure, pulse pressure, total cholesterol, triglycerides, plasma fibrinogen, uric acid, creatinine and blood urea nitrogen were analyzed by Mann-Whitney U-test while the group did not follow the normal distribution, and were presented as median (IQR); LDL-C was examined by Welch test while the homogeneity test was not homogeneity.

uric acid, and blood urea nitrogen were significantly higher ($p < 0.001$), and HDL-C lower ($p < 0.001$) in the CAD-RADS score 3-5 group compared to the CAD-RADS score 0-2 group (Table 1). There were significant differences in hyperlipidemia, triglycerides, and serum creatinine between the two groups. However, our results did not reveal any association between diastolic blood pressure ($p = 0.052$), total cholesterol ($p = 0.265$), LDL-C ($p = 0.572$) and different CAD-RADS scores. Table 1 illustrated the univariate analysis for the association between cardiovascular risk factors and CAD classified using CAD-RADS.

## Model evaluation

We applied six ML algorithms to the test dataset. Table 2 compared the predictive values of different models regarding their AUC, accuracy, sensitivity, and specificity. All the models demonstrated good performance (AUC>0.6) in predicting CAD-RADS scores. RF and LDA models showed excellent discrimination with an AUC of 0.832 and 0.81, respectively. SVM and NN had an acceptable performance, and DTC showed the lowest discriminatory

**Table 2   Evaluation of ML models for predicting CAD-RADS scores.**

|  | Sensitivity | Specificity | Accuracy | AUC |
|---|---|---|---|---|
| RF | 0.762 | 0.762 | 0.773 | 0.832 |
| SVM | 0.772 | 0.772 | 0.773 | 0.772 |
| KNN | 0.624 | 0.616 | 0.636 | 0.707 |
| LDA | 0.715 | 0.714 | 0.716 | 0.810 |
| NN | 0.750 | 0.750 | 0.750 | 0.773 |
| DTC | 0.682 | 0.682 | 0.682 | 0.682 |

**Notes.**

AUC, area under the receiver operating characteristic curve; RF, Random Forest; SVM, Support Vector Machine; KNN, K-Nearest Neighbors; LDA, Linear Discriminant Analysis; NN, Neural Network; DTC, Decision Tree Classification. Analysis was done with k-fold crossvalidation ($k = 10$). Sensitivity: (TP)/(TP + FN), Specificity: (TN)/(TN + FP), Accuracy: (TP + TN)/(TP + TN + FP + FN).

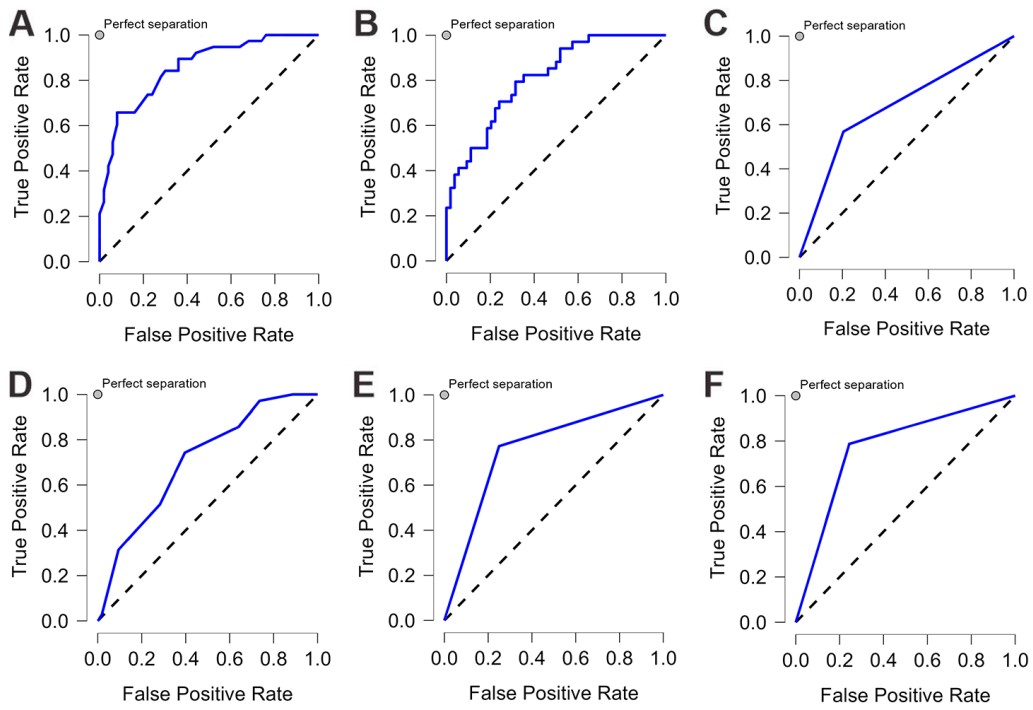

**Figure 3   The ROC curve of each machine learning model.** Each machine learning model was assessed by the ROC curve, which plots a curve according to its true positive rate ($y$-axis) against its false positive rate ($x$-axis). The larger area under the curve, the better the prediction accuracy of the model. (A) Random forest (RF). (B) Linear discriminant analysis (LDA). (C) Decision tree classification (DTC). (D) k-nearest neighbors (KNN). (E) Neural network (NN). (F) Support vector machine (SVM).

ability with an AUC of 0.682. After tuning for the threshold, the SVM model achieved the highest sensitivity and specificity, both at 0.772. Both RF and SVM showed the highest accuracy, both at 0.773. Figure 3 illustrated the ROC-AUC for the six models.

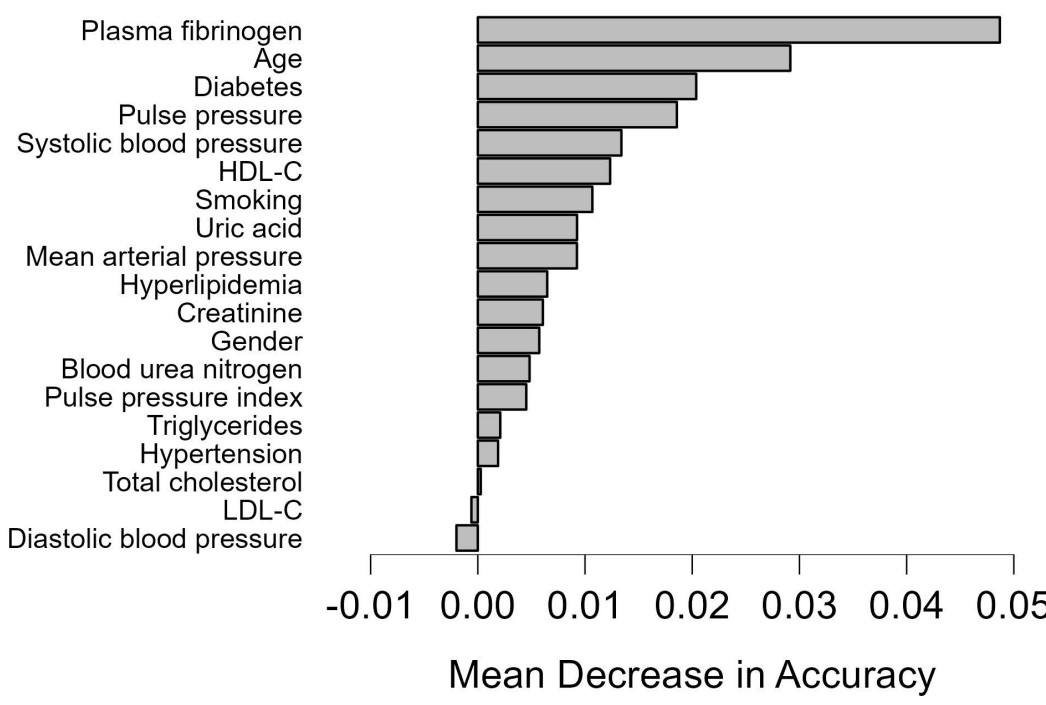

**Figure 4** **Feature importance based on the random forest model.** Feature importance based on the random forest model. HDL-C, high-density lipoprotein cholesterol; LDL-C, low-density lipoprotein cholesterol.

## Result of feature importance

We employed the RF prediction model to rank all features based on their significance in test data, using k-fold cross-validation ($k = 10$). Figure 4 showed the order of features for model development. Nineteen features were chosen for predicting CAD- RADS scores. The feature importance analysis revealed that plasma fibrinogen was the most important feature for the classification task, followed by age and DM (Fig. 4).

## DISCUSSION

CAD is a serious disease that affects both health and function. Identifying risk factors is crucial for preventing acute coronary events in CAD patients. While many risk factors have been proposed for CAD patients, few studies have investigated risk factors associated with CAD-RADS classification, and no effective systematic model has been proposed to predict whether a patient is at high risk of coronary heart disease. Previous studies investigating the application of AI in the diagnostic pathway of CAD have used different AI algorithms (*Khalaji et al., 2022*; *Muscogiuri et al., 2020*; *Shariatnia et al., 2022*). In this study, we investigated whether CAD-RADS scores could be predicted using ML algorithms based on risk factors data from CAD patients. Our cohort study utilized the cardiovascular medicine databank from two clinical research institutes, which contained diverse demographic information and can provide reliable data on patients with CAD.
CAD-RADS, as a powerful standardized reporting tool, may facilitate further research and provide a framework for standardized collection of CCTA reports across multiple sites for quality improvement and benchmarking (*Cury et al., 2016*). In CCTA interpretation, a proper assessment of the CAD extent, severity, and characteristics largely depends on the reader's clinical skills and experience. Despite proper CAD assessment, even experienced readers might misclassify cases due to a lack of knowledge of the CAD-RADS classification (*Foldyna et al., 2018*). In our study, images were post-processed using AI software. In the future, automated classification systems may combine image analysis and standardized reporting tools, leading to more reliable and faster CAD-RADS assessment (*Foldyna et al., 2018*), especially for CAD-RADS 3-5 score CAD patients, a CAD-RADS grade of 3 or greater suggests consideration of functional evaluation and anti-ischemic or preventative drugs (*Cury et al., 2016*; *Huang et al., 2020*; *Muscogiuri et al., 2020*; *Rubinshtein & Hamdan, 2020*).

Multiple ML algorithms can be utilized for feature importance analysis, with RF being a commonly employed method, RF can ameliorate prediction accuracy without considerably increasing the calculation amount, maintain high predictive performance, is a very effective method in feature screening and classification . In a recent research, RF model illustrated a good AUC of 0.948 to identify CAD patients from controls, which exhibited favorable predictive capability and clinical application value (*Wang et al., 2021*). In agreement with this finding, another study compared various ML models for estimating the diagnosis of CAD, and their results showed that RF predictive model achieved 92.04% accuracy and 92.20% ROC respectively and was identified as the best model among other models (*Muhammad et al., 2021*). The utilization of RF in CAD had been highlighted in other literature as well (*Liu et al., 2021*; *Saharan et al., 2021*). Our research has demonstrated that RF exhibited exceptional predictive performance, with an AUC of 0.832, surpassing other models in comparison. These findings are consistent with existing literatures on the potential applications of RF. Our findings also indicated that LDA model demonstrated a comparable predictive ability to the RF model, with an AUC of 0.81. LDA has been recommended as a predictive model with excellent accuracy, sensitivity, and specificity in the applications in cardiovascular diseases (*Ricciardi et al., 2020*; *Shariatnia et al., 2022*).

Plasma fibrinogen, as a coagulation index, was the most important feature based on our feature selector, and it was independently associated with coronary severity and complexity in patients with CAD. Plasma fibrinogen, a marker of inflammation and coagulation, may stimulate coagulation, platelet aggregation, and vascular endothelial dysfunction, mediate the transportation of adhesion molecules on the surface of the endothelium and their further migration to the intima, trigger proliferation and migration of smooth-muscle cells to increase coronary plaque vulnerability (*Loukas et al., 2002*; *Song et al., 2015*; *Tabakcı et al., 2017*), and is a potentially suitable target for CAD. Many studies have examined the role of plasma fibrinogen levels alone in the prediction of CAD events. Song et al. reported that the plasma fibrinogen levels of CAD patients were 0.94-fold higher than the control group and showed a significant association between plasma fibrinogen level and CAD risk (*Song et al., 2015*). A meta-analysis confirmed that an increase in fibrinogen concentration by 1 g/L, depending on age and sex differences, was associated with a

 

higher risk of CAD by 2.42 (*Danesh et al., 2005*). Gąsior's study (*Gasior et al., 2018*) showed that for patients with non-critical stenosis in coronary arteries, higher plasma fibrinogen concentration would predispose them to the occurrence of cardiovascular events, plasma fibrinogen was proved to be a parameter related to the frequency of revascularization. In a recent community-based cohort study (*Hsieh et al., 2022*), a total of 2,222 participants who underwent plasma fibrinogen measurements and did not have CVD at baseline were recruited in the Taiwanese population. Their findings showed that participants with higher fibrinogen levels tended to have a higher risk of CAD, indicating that a high level of fibrinogen may be a risk factor for CAD. These observations indicated that plasma fibrinogen is independently associated with coronary severity and complexity in patients with CAD. In agreement with these studies, our model suggested that plasma fibrinogen was a significant risk factor for high CAD-RADS scores, patients in CAD-RADS score 3–5 group exhibited higher plasma fibrinogen levels than the CAD-RADS score 0–2 group ($p < 0.001$). These findings highlight the potential benefits of monitoring blood plasma fibrinogen concentrations in preventing CAD.

In our study, age was the second most important risk predictor for CAD-RADS scores. Similar findings have been reported that age was the second most important risk factor for 5-year mortality in CAD patients undergoing PCI (*Liu et al., 2021*). *Sun et al. (2012)* demonstrated that the percentage of patients with significant coronary artery stenosis increased to 38% in patients aged over 65 years compared to less than 15% in patients under 56 years. *Kim et al. (2021)* used the RF model to define the relative importance of age on coronary plaque progression, they found that the rate of whole-heart plaque progression and dense calcification increases depending on age, as important as any other traditional cardiovascular risk factors. Our research demonstrated that DM was also a significant risk factor for high CAD-RADS scores. Similar to our finding, a previous study showed that patients with DM demonstrated more obstructive CAD on CCTA than patients without DM (*Van den Hoogen et al., 2020*). Both age and DM play crucial roles in plaque growth and the progression of coronary atherosclerosis, as evidenced by the size, volume, and density of coronary atherosclerotic plaque which directly impact the degree of stenosis in the coronary artery lumen, thereby affecting CAD-RADS scores.

In summary, our findings suggested that elevated plasma fibrinogen levels, advanced age, and DM were significant predictors of CAD-RADS 3–5 scores. Monitoring plasma fibrinogen and blood glucose levels may offer additional information for the prevention of CAD in clinical practice. Individuals with elevated level of plasma fibrinogen, blood glucose or advanced age should receive increased attention in the CAD prevention efforts. In addition to plasma fibrinogen, age and DM, other variables such as pulse pressure, HDL-C, pulse pressure index, mean arterial pressure, SBP and smoking are also relatively significant in stratifying a patient's risk for CAD. These risk factors seldom occur independently but rather tend to cluster together with other cardiovascular risk factors. Modifying these risk factors may be effective in preventing CAD progression and reducing CAD-RADS scores. Further research with a larger sample size of Chinese patients with CAD is necessary to provide more conclusive evidence regarding these associations.

## LIMITATIONS

This study has both strengths and limitations: Firstly, there are still some disadvantages to using ML in cardiovascular practice. There are many unmeasured or unknown important variables, and different classifiers for the same dataset may not all be equally robust (*Chuah et al., 2022*). Data availability also limits the generalizability of ML algorithms. The data used for training ML models are typically acquired from one or several laboratories, health centers, or hospitals (*Shu, Ren & Song, 2021*), the outcomes may vary among diverse populations, as a result, external validation of the models is required. Secondly, the data were collected from two hospitals, the study population was relatively small. Another limitation was the absence of pathological confirmation of CAD severity and the cross-sectional design. Moreover, the data were collected retrospectively, which may lower the reliability of evidence compared to prospectively collected data. Lastly, we only considered 19 traditional risk factors for CAD. Future studies should include more variables to further validate our findings.

## CONCLUSION

This study indicated that RF outperformed other models in predicting CAD-RADS scores among CAD patients, making it a recommended predictive model for identifying high-risk patients with CAD-RADS 3–5 scores. The most significant feature selection were plasma fibrinogen, age and DM, indicating that combined strategies targeting these factors may be effective in preventing the burden of CAD. We hope this study can serve as a valuable resource for future research on this topic.

### Funding

This research was supported by the Guidance Project of Hengyang City (No. 202222035663), the Scientific Research Project of Hunan Provincial Health Commission of China (No.202109011222; No.202203013975), the Natural Science Foundation of Hunan Province (2022JJ40386), and the Scientifc Research Project of Hunan Provincial Department of Education (No. 21B0407). The funders had no role in study design, data collection and analysis, decision to publish, or preparation of the manuscript.

### Grant Disclosures

The following grant information was disclosed by the authors:
The Guidance Project of Hengyang City: 202222035663.
The Scientific Research Project of Hunan Provincial Health Commission of China: 202109011222, 202203013975.
The Natural Science Foundation of Hunan Province: 2022JJ40386.
The Scientifc Research Project of Hunan Provincial Department of Education: 21B0407.

### Competing Interests

The authors declare that there are no competing interests.

## Author Contributions

- Yueli Dai performed the experiments, analyzed the data, prepared figures and/or tables, authored or reviewed drafts of the article, and approved the final draft.
- Chenyu Ouyang performed the experiments, analyzed the data, prepared figures and/or tables, authored or reviewed drafts of the article, and approved the final draft.
- Guanghua Luo conceived and designed the experiments, authored or reviewed drafts of the article, supervised the project, and approved the final draft.
- Yi Cao analyzed the data, prepared figures and/or tables, and approved the final draft.
- Jianchun Peng analyzed the data, authored or reviewed drafts of the article, supervised the project, and approved the final draft.
- Anbo Gao conceived and designed the experiments, performed the experiments, authored or reviewed drafts of the article, and approved the final draft.
- Hong Zhou conceived and designed the experiments, analyzed the data, prepared figures and/or tables, and approved the final draft.

## Human Ethics

The following information was supplied relating to ethical approvals (i.e., approving body and any reference numbers):

The study was approved by the human ethics review board of University of South China (2022020587).

## Data Deposition

The raw measurements are available in the Supplementary File.

## Supplemental Information

Supplemental information for this article can be found online at http://dx.doi.org/10.7717/peerj.15797#supplemental-information.

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
