# Peer review of "Risk factors for high CAD-RADS scoring in CAD patients revealed by machine learning methods: a retrospective study"

_PeerJ, doi:10.7717/peerj.15797_

## Round 0.1 · original submission · Major Revisions

All the reviewers raised major concerns. The authors should carefully address them.

Reviewer 1 ·

Basic reporting

1.Which group was mainly studied? In the abstract ,in line 38-39, “…the CAD-RADS was stratified into 2 groups: no to minimal CAD (CAD-RADS 0-2), moderate and severe CAD (CAD-RADS 3-5) ”, in line 46-47, (52.9%) subjects were CAD-RADS score 0-2 group and 208 (47.1%) subjects were CAD-RADS score 3-5 group .The group descriptions should be consistent.Why divide the subjects into two groups instead of four?
2.There are some blank spaces in the manuscript (line114 in Materials & Methods; line 3 in table2)
3.In line 153-155, “After acquisition, the images were processed using artificial intelligence (AI) software (Shukun, Beijing) ”. Please provide the software version

Experimental design

1.There are unclear descriptions of in the manuscript. For instance, in line 32, “…machine learning and deep learning algorithms”; line 57, “Machine learning algorithms…”,line 90-91, “machine learning and deep learning…”;line108, “…AI-based algorithms…”; line 111, “…artificial intelligence-based algorithms”. Description is inconsistent, which method was mainly studied? Similar issues can be found in many places in the manuscript. Please carefully rephrase these descriptions.
2.Please state your hypothesis and object in detail in the last paragraph of the introduction.
3. In the discussion section, it seems that there are too many defects in ML, and there are also too many articles inconsistent with the author's research conclusions.

Validity of the findings

no comment

Additional comments

no comment

Reviewer 2 ·

Basic reporting

In this paper, the prediction models of CAD-RADS were first constructed, and then the risk factors leading to severe vascular stenosis were revealed through the feature-importance analysis of the models. The research method and design are novel and the conclusion is reliable. But, some descriptions in this paper are irregular and easy to cause ambiguity. For example, the description of the prediction models are inconsistent (Line40, 111,232,242), whether it is the prediction of risk factors or the prediction of CAD-RADS classification; I think it should be the latter. Methodological description is not clear and detailed. The discussion part needs to be revised. The significance of this study and the clinical value of finding these risk factors need to be further explained in the discussion. There are many grammatical errors and the language expression needs to be further improved.

Experimental design

Methods and results

Line166-168 :It can be put into the discussion;
Line176: The expression of “feature selection algorithm” is not accurate. How exactly do “the top features ” get? It should be further stated;
Line154: The specific name and version of the AI software should be specified;
Line156: Rating of CAD-RADS score. It is not stated whether the RADS score is obtained by AI software or human evaluation?
Line238:It is not feature selection, but the result of feature importance (or risk factor) analysis.

Validity of the findings

Line239: The LAD model also shows excellent discrimination against severe CAD, so the risk factors reflected by the LAD model should also be analyzed, and the results of the LAD model can be compared with those of the RF model, and the differences between the results of the two models should be analyzed;
Line242,244: At least 3 or more risk factors that ultimately lead to high CAD scores should be concluded. Whether there is a strong correlation among them, whether this correlation has any influence on the final conclusion, and the clinical guiding significance of the conclusions should be explained in the discussion.

Additional comments

1. The title does not summarize the topic of the article, and it is suggested to be corrected, such as: "Risk factors for high CAD-RADS scoring in CAD patients revealed by machine learning methods".
2. Abstract:
Line39,40: "Cardiovascular risk factors were predicted with..." Such an expression is inaccurate and leads to ambiguity. The study did not use these models to predict risk factors; Instead, risk factors are used to predict the two CAD-Rads classification, and then feature importance analysis is carried out on the model to find out the risk factors that play a major role in CAD classification.
3. Language and format
Syntax error: Line267,268, an artificial intelligence (AI) software
Typography problems: both ends are aligned;
Punctuation: Spaces should be added before and after all brackets (Line 40,66,72,90...). ;
A comma should be followed by a space (Line 41:,Decision);
There is no space before or after the equal sign, and the format should be uniform (Line 55, 226, 227...). ;
Abbreviations can be used directly after their first appearance (Line 206, 207,183, 184);
Numbers should start sentences in full (Line 216:142, one hundred and forty-two);
P values should be italicized (Line 222).
4. Discussion :
Line 245: Discussion;
Line259,260: Delete “based on our findings”;
Line278,279: Inaccurate expression, RF can be used for feature importance analysis, but is not a feature selection method;
Line278-308, 309-324: The author introduces a lot about the application of RF/LAD algorithm in CAD diagnosis, which has no great significance and should be simplified;
Line375: Punctuation error;
The discussion should focus on explaining the reliability of the results, the innovation of the study, and the contribution to improving clinical decision-making in CAD, rather than blindly reviewing and comparing the previous literature.
5. Conclusions: It needs to be more refined.
6: Figures
Figure 3. Why are there two curves in ROC? There should be only one result for each model.
Figure 4. The feature importance analysis results of the LAD model should be increased.

·

Basic reporting

1. In line 32, “This study aimed to investigate a variety of machine learning and deep learning algorithms…”, this study was focused on machine learning or deep learning algorithms? It is suggested that the author specify the main study method, and unified throughout the manuscript.

2. In line 50, “…HDL-C lower (1.246±0.355 VS 1.366±0.378, p < 0.001)”. The numbers are verbose, HDL-C lower (p < 0.001) is more clearly and concisely. These similar descriptions of numbers also exist in the results and discussion section, please carefully check these descriptions.

3. There are some blank spaces in the manuscript (line114 in Materials & Methods; line 3 in table2).

4. Descriptive confusion. Eg, In line 42, “Linear discriminant analysis(LDA)”. In line 54 and 59, “Linear Discriminant analysis ”, in line 105 “discriminant analysis”,and in table 2, “Linear Discriminant” ,the descriptions should be consistent, similar issues can be found in many places in the manuscript. Eg, in line 45, “ CCTA examinations” is inconsistent with “CTA examination”in line 60 ,eg, ML and RF, please carefully rephrase these problems.

5. Please check all references and read the abstract or even the full text of the reference to ensure the correct reference.

6. In line 222, “…and blood urea nitrogen were significantly higher (p < 0.001),”. “p” should be shown in italics. Besides, the “p-value” applied in the manuscript were inconsistent. For example, “p-value” in line 204, different from “p value” in Table 1. Therefore, the manuscript needs another run of proofreading for typos and style.

7. When abbreviation was used initially, it should follow the full word in parentheses, for instance, MLAs.

8. Please pay attention to tense expressions, eg, in line108, “although some literature has applied AI-based algorithms for the prediction”, should be modified “some literatures had”. And in line 109, “few studies on CAD-RADS scores had been evaluated to predict risk”.

9.Please check the format of article, eg. Line 61, subjects; line 246, conclusion. line 430-436, acknowledgement.

10.line 109-112, 252-255 the expressions were contrast to the experiment design, please check throughout the manuscript.

11. There are too many abbreviations in the abstract. It is recommended not to define abbreviations that occur less than three times.

12.In 426-428, “The integration of this model into the hospital's software environment could enable continuous monitoring of at-risk CAD patients, assist cardiologists in earlier CAD diagnosis, and achieve the goal of precision medicine”, how did the author come to this conclusion? Did the author do any clinical studies?

14. Abstract and conclusions is too long, although the logic is clear. Please simplify.

Experimental design

1.The time interval between collection clinical risk factors and CCTA data should be specified as it may substantially affect results.

2. Line 116, Place where the study was carried out should be stated clearly.

3.The CCTA data were collected with different scanners,are the scanning parameters were the same? If not, how do you balance this factor?

4. Statistical Analysis section. The authors should state clearly what did they want to compare.

5.In line 45, “A total of 442 patients with CCTA examinations were included in this study”. Please specify the patients.

Validity of the findings

The current study lack of external validation, however, these data were collected in two sites, why not use data of one site to validate the main results?

Additional comments

None

---

## Round 0.2 · Minor Revisions

There are still several concerns that should be carefully answered.

Reviewer 1 ·

Basic reporting

Clear and professional English was used in this article. The structure of the article is clear.

Experimental design

Experimental design is scientific and reasonable.

Validity of the findings

The results are credible.

Reviewer 2 ·

Basic reporting

Some paragraph indentation format and punctuation formats (Line87,102,246,257,267,277,296)are inconsistent,please check again.

Experimental design

Generally speaking, CNN is not good at processing structured data. In this article, a CNN model was used for constructing prediction model. I would like to know which kind of CNN model was used. Please provide a brief description in the methods section.

Validity of the findings

Please check the ROC in Figure 3,there should be only one curve for each model.

Additional comments

none.

·

Basic reporting

1. There are too many keywords.
2. The logic of discussion is not clearly enough. Eg. line 253-260 is more suitable to be put in the limitation. Line 301-303, it is more appropriate to put forwards the summarized findings, follow by elaborate explanation.

Experimental design

No comment.

Validity of the findings

No comment.

Additional comments

None

---

## Round 0.3 · accepted · Accept

The manuscript can be accepted now!